# Revisiting the Rotational Field TMS Method for Neurostimulation

**DOI:** 10.3390/jcm12030983

**Published:** 2023-01-27

**Authors:** Yiftach Roth, Samuel Zibman, Gaby S. Pell, Abraham Zangen, Aron Tendler

**Affiliations:** 1Brainsway Ltd., Jerusalem 9777518, Israel; 2Department of Life Sciences, Ben Gurion University of the Negev, Beer Sheva 8410501, Israel

**Keywords:** transcranial magnetic stimulation, rotational field, TMS 360°, neurostimulation, orientational sensitivity, depolarization

## Abstract

Transcranial magnetic stimulation (TMS) is a non-invasive technique that has shown high efficacy in the treatment of major depressive disorder (MDD) and is increasingly utilized for various neuropsychiatric disorders. However, conventional TMS is limited to activating only a small fraction of neurons that have components parallel to the induced electric field. This likely contributes to the significant variability observed in clinical outcomes. A novel method termed rotational field TMS (rfTMS or TMS 360°) enables the activation of a greater number of neurons by reducing the sensitivity to orientation. Recruitment of a larger number of neurons offers the potential to enhance efficacy and reduce variability in the treatment of clinical indications for which neuronal recruitment and organization may play a significant role, such as MDD and stroke. The potential of the method remains to be validated in clinical trials. Here, we revisit and describe in detail the rfTMS method, its principles, mode of operation, effects on the brain, and potential benefits for clinical TMS.

## 1. Introduction

Repetitive transcranial magnetic stimulation (rTMS) is a safe [1], effective [2], and non-invasive treatment for treatment-resistant MDD with high cost-effectiveness compared to pharmacotherapy [3]. Double-blind placebo-controlled (DBPC) multicenter trials [4,5,6,7,8] resulted in US Food and Drug Administration (FDA) clearance for the treatment of major depressive disorder (MDD) with figure-8 or H1 TMS coils, for the treatment of obsessive-compulsive disorder (OCD) with the H7 Coil, and for the treatment of smoking cessation with the H4 Coil. The figure-8 coil and the H-coils crucially differ in generated electric fields, such that the figure-8 induces a focal and superficial electric field and stimulates the focal superficial cortical region underneath the coil, while the H-coils induce deeper and broader penetration of electromagnetic stimulation into the brain [9].

TMS is performed by passing a transient electric current pulse through a coil placed on the scalp. This current generates a temporary electric field that passes through the scalp and skull activating neurons in the underlying brain tissue [10]. Repetitive application of TMS (rTMS) may induce long-term neuroplastic changes in the excitability and connectivity of relevant brain circuits. This is believed to underlie its use as a therapeutic intervention for various neuropsychiatric indications [11,12].

For TMS to reach its full potential, the field of clinical TMS must overcome the challenge of substantial inter-individual variability of clinical outcomes. This high variability stems from a combination of many factors, including demographic, genetic, and neurophysiological characteristics [13], the baseline brain state [14], and the exact timing of stimulation relative to endogenous brain activity [15].

An additional but often underestimated contribution to this variability is the orientational sensitivity of TMS. The electric field is a vector quantity, such that at any specific time and location, the electric field has a specific amplitude and direction, which is the sum of all the sources of the electric field. In the case of TMS, for maximal induced electric field [16] with its direction parallel to the neuron’s somato-dendritic axis, simulation has shown that neural stimulation is initiated at axon terminations or at bend points (where the axon bends away from the direction of the TMS electric field) [17,18,19,20,21,22]. Moreover, the polarity of the field along the nerve axis plays a crucial role in the effectiveness of the stimulation. For a certain polarity, membrane depolarization of sufficient intensity would occur at the bend point and lead to neural stimulation. On the other hand, the opposite polarity would lead to membrane hyperpolarization, reducing the chance for stimulation. This is most readily demonstrated by the differences in hand motor threshold between anterior–posterior and posterior–anterior-directed TMS [19,20,22]. Hence, neurons with a structure parallel to the induced electric field are most efficiently stimulated. However, neurons in the brain are in general arranged in various orientations, such that only a small proportion of the neural population is stimulated by TMS. This inherent limitation of unidirectional TMS may partially explain the observed variability in the magnitude of excitability modulation, which can manifest even as a difference in the sign of the effect over a population [23], as well as contributing to variability in clinical outcomes.

Current clinical practice follows the same one-size-fits-all approach of orienting the coil 45° with respect to the sagittal plane, as was described over 20 years ago in the relatively early days of the technology [24]. The orientation was chosen based on work on the hand knob that indicated this to be optimal for stimulation of a motor response over a population, notwithstanding evidence of inter-individual variability [25,26]. However, the dorsolateral prefrontal cortex (dlPFC), which is the primary target in the treatment of depression, has greater anatomical and functional complexity [27]. Consequently, the optimal coil orientation for effective treatment is expected to vary between individuals.

The emerging approach of rotational field TMS (rfTMS), also termed TMS 360°, has been presented [28] and later tested in human subjects [29,30] in which two orthogonal coils are operated with a 90° phase shift between them. The combined electric field vector of rfTMS from both coils rotates up to a complete cycle during the TMS pulse. This method overcomes the inherent limitation of conventional unidirectional TMS by influencing neurons in various orientations both in scenarios in which the optimal orientation is unknown or in which an optimal orientation does not exist. This unique advance bears enormous potential for therapeutic applications in various neuropsychiatric disorders.

In this Perspectives article, we explain the principles of conventional TMS and the limitations of unidirectional TMS. We give a detailed presentation and visualization of the rfTMS method and its potential implications on clinical applications and neuroscience in general. In addition, we revisited the results of rfTMS in healthy subjects and performed analyses that shed light on the interplay between orientational sensitivity and depth in TMS.

## 2. TMS Principles

In a standard TMS exam, the subject’s resting motor threshold (rMT) is determined using single pulses. The TMS coil is placed over the motor cortex of the relevant muscle (in the hand or foot). The position and orientation of the coil are varied while single TMS pulses are administered, until the optimal spot on the scalp for stimulation of the motor cortex is localized and the rMT is defined. The rMT is the minimal stimulator power output required to elicit a motor response. After rMT determination, the coil is moved to the treatment position (i.e., the left dlPFC in most treatments of MDD), the coil is attached to the head, and a treatment session of rTMS is administered. Currently available TMS devices produce biphasic pulses. The biphasic pulse of electrical current produced by the TMS device is brief with a duration of less than 1 millisecond and consists of a positive peak during which the current travels in one direction followed by a negative one during which the current travels in the reverse direction (Figure 1).

The induced electric field lines are parallel to the current in the coil (Figure 2I).

For example, in a figure-8 coil, the current flows in circles in the two wings. The lines of the electric field induced in the brain tissue underneath are parallel to the current. The strongest field is induced under the figure-8 coil central segment, and its direction would be parallel to the coil axis (Figure 2II). At any point in the brain, there is an ensemble of neurons in various orientations (Figure 2II). Only neurons aligned parallel to the induced electric field will be stimulated, while neurons in other orientations will not be stimulated.

More accurately, as stated above, stimulation is usually initiated at bend points or axon terminals.

The scenario of stimulation initiation at a bend point is shown in Figure 2III, for a neuron under the coil center where the electric field is maximal and the axon is parallel to the field and bends away from it (Figure 2IIIa,b). The induced field leads to transmembrane potential V_m_ across the voltage-gated ion channels (Figure 2IIIc) inducing depolarization. Above a certain threshold, this depolarization leads to the opening of the ion channels, inflow of sodium ions into the intracellular space, and initiation of action potential.

The scenario of initiation of stimulation at an axon terminal is shown in Figure 2IV for a neuron under the coil center where the electric field is maximal and the axon is parallel to the field and terminates at a synapse (Figure 2IVa,b). The induced field leads to a swing of transmembrane potential V_m_ across the voltage-gated ion channels (Figure 2IVc), leading to depolarization. Above a certain threshold, this depolarization leads to the opening of the ion channels, inflow of sodium ions into the intracellular space, and initiation of action potential.

## 3. Rotational Field TMS (rfTMS) Method

For this Perspectives article, we surveyed publications related to the rotating field TMS method, as well as all publications related to the roles of orientation in TMS. The basic idea in rfTMS is to place two coils perpendicular to each other, one on top of the other, over the head (Figure 3a), and to operate them not simultaneously, but with a phase lag of 90°, namely, ¼ of a cycle.

In the example of Figure 3, the lower (red) coil is an H7 Coil, and the upper (green) coil is perpendicular to the H7 Coil and has the same inductance. In general, the coil inductance affects the pulse duration. To induce a circularly rotating electric field, the two coils need to have matched inductances so that their pulse duration are identical. The two coils induce a maximal electric field at the same point in the brain, underneath both coils’ central segments. In Figure 3, the lower (red) coil induces an electric field along the anterior–posterior axis and the upper (green) coil induces an electric field along the lateral–medial (left-right) axis.

If only the lower coil is operated, only neurons aligned along the anterior–posterior axis are optimally stimulated (Figure 3b). If only the upper coil is operated, only neurons aligned along the lateral–medial axis are optimally stimulated (Figure 3c). If the two coils are operated simultaneously, the induced electric field would be the combination of the effects of the two coils. If the amplitudes are identical, the induced electric field would be at 45° to the lateral–medial axis (Figure 3d).

Hence, the idea of rfTMS is to operate the two coils not simultaneously, but instead with a lag of ¼ of a cycle (equivalent to a phase shift π/2); Figure 4 demonstrates this. The upper (green) coil is operated with a lag of 90° degrees (¼ cycle) after the lower (red) coil. The pulse timing diagram is shown on the right of each panel of Figure 4, presenting the electric field pulse induced by each coil. The pulses have a Cosine shape, and a full cycle (360° degrees) is completed when the amplitude re-assumes its initial value. If both coils are operated with biphasic pulses, then the lower (red) coil is terminated at t = 2 π. In such a case, the field vector would cover only 270° of the phase space. Hence, in the demonstration shown in Figure 4, the lower (red) coil is operated with two consecutive biphasic pulses, in order to have an overlap with the upper (green) coil between t = 2 π and t = 2.5 π so that the field vector completes a full cycle of 360°.

The field of the lower coil is induced along the posterior–anterior axis. After ¼ of a cycle (90° degrees or π/2), the amplitude of the electric field induced by the lower (red) coil reaches zero. At that instant, the upper (green) coil is operated and induces a field along the left–right axis.

In the following, we will go step by step during the pulse and see how exactly the combined effects of the two coils lead to circular rotation of the induced electric field direction. Figure 4 presents the evolution of the field during the pulse. In Figure 4a top left is shown the rfTMS coils array with the lower red coil and the upper green coil in orthogonal orientations. The timing in the pulse is indicated by circles on the pulses of the lower (red) coil and the upper (green) coil in the pulse diagram shown for each stage. The white arrow on the colored field map over the brain indicates the field direction at the point of the maximal field. At the pulse onset (Figure 4a; time t = 0), only the lower coil is operated, and it induces a field in a posterior–anterior direction. At time t = 0.4 π (Figure 4b; t = 0.4 × 180° = 72°), still, only the lower coil is operated, and the field is still in a posterior–anterior direction but with reduced intensity. At time t = 0.5 π (Figure 4c; t = 0.5 × 180° = 90°), the field induced by the lower coil is zero, and the upper coil is now operated and induces a field in a left–right direction. At time t = 0.7 π (Figure 4d; t = 0.7 × 180° = 126°), the upper coil still induces a field in the left–right direction with reduced intensity (green arrow over the brain), yet the lower coil now induces a field in a direction opposite to its initial direction, in the anterior–posterior direction (red arrow over the brain). The resultant field is the vector sum of the fields induced by the two coils. Hence, the field that will be induced at that instant will be at an angle downward from the left–right axis, as indicated by the white arrow. It is important to understand that at each instant, there is a field in one and only one direction since the electric field is a vector. Hence, at that instant and point, there is no field along the anterior–posterior axis nor along the left–right axis, but only in the direction indicated by the white arrow. At time t = 1.0 π (Figure 4e; t = 180°), the field induced by the upper coil is zero, while the lower coil induces a field at maximal amplitude in the anterior–posterior direction. At time t = 1.25 π (Figure 4f; t = 1.25 × 180° = 225°), the amplitude of the field induced by the lower coil is reduced while the upper coil now induces a field in a direction opposite to its initial direction, in the right–left direction. Hence, the field vector keeps rotating and now points in the direction indicated by the white arrow, to the left and posteriorly. At time t = 1.5 π (Figure 4g; t = 1.5 × 180° = 270°), the field of the lower coil is zero and the upper coil induces a field at maximal amplitude in the right–left direction. At time t = 1.75 π (Figure 4h; t = 1.75 × 180° = 315°), the amplitude of the field induced by the upper coil is reduced while the lower coil now induces a field in the posterior–anterior direction. Hence, the field vector keeps rotating and now points in the direction indicated by the white arrow, to the left and anteriorly. At time t = 2 π (Figure 4i; t = 2 × 180° = 360°), the field induced by the upper coil is zero, while the lower coil induces a field at maximal amplitude in the posterior–anterior direction. Up until now, the field vector spanned all the directions from left–right to posterior–anterior in a clockwise rotation. To span the remaining directions, at time t = 2.25 π (Figure 4j; t = 2.25 × 180° = 405°), the amplitude of the field induced by the lower coil is reduced while the upper coil now induces a field in the left–right direction. Hence, the field vector keeps rotating and now points in the direction indicated by the white arrow, to the right and anteriorly. At time t = 2.5 π (Figure 4k; t = 2.5 × 180° = 450°), the field of the lower coil is zero and the upper coil induces a field at maximal amplitude in the left–right direction. Hence, now the field vector has spanned all directions. The result is that neurons oriented in various directions are all stimulated within less than a millisecond (Figure 4k), in contrast to conventional unidirectional TMS where only a small fraction of the neural population, oriented parallel to the induced field, are stimulated when operating close to the threshold of neural stimulation (as in the examples of Figure 3). Increasing, the intensity above the threshold would recruit additional neurons with components aligned at angles close to the field. However, since clinical applications are generally limited to 120% of the motor threshold [1], stimulation with unidirectional TMS is limited to structures with favorable orientations.

It is instructive to see the field as experienced by a certain neuron having a bend point or axon terminal in a specific orientation. At any instant of the pulse, the field induced at the site in a specific direction is the vector sum of the contributions of the two coils. Figure 5a presents the field along the posterior–anterior (P-A) axis in such a site. The lower coil induces two biphasic pulses as stated above. The upper coil has no contribution along the P-A axis. The threshold for neural stimulation is set at 0.9. The red circle on the neuron and on the graph indicates the spatial site as well as the time point of action potential initiation, respectively.

Figure 5b presents a site with preferred orientation in the left–right direction. In this case, the only contribution is from the upper coil, and the action potential is initiated at the time point of 0.5 π after the upper coil is operated.

A more complex case is presented in Figure 5c, depicting a neuron with a preferred orientation of 0.7 π. Here, both coils contribute by the projections of their fields on the respective axis. Between t = 0 and t = 0.5 π, only the lower coil is operated and induces field in the opposite direction, which leads to hyperpolarization. Action potential is initiated at about t = 0.6 π, after the upper coil is operated at t = 0.5 π, and the field at the site in the preferred direction leads to depolarization and reaches the threshold. The essential prerequisite for neural stimulation is not the particular pulse shape (biphasic, monophasic, or other) but the induction of depolarization above the threshold, as demonstrated in the controllable TMS (cTMS) setup, e.g., ref. [31], with near-rectangular electric field pulses.

An animation visualizing the rotational field TMS effect is added in the Appendix A.

A demonstration of the orientational dependence of action potential initiation induced by TMS was presented in a recent modeling study [22]. As can be seen in Figure 6, initiation of action potential by monophasic posterior–anterior (green) or anterior–posterior (magenta) stimulation depends on both the orientation and directionality of the axons.

## 4. Re-Visiting the rfTMS Clinical Results

The physiological effects of rfTMS were compared with conventional unidirectional TMS (udTMS) in the motor cortex of healthy subjects [29,30]. rfTMS induced a significantly lower resting motor threshold (rMT) in both the hand and leg motor cortices compared to unidirectional H7 Coil Deep TMS [29,30]. In accordance with previous studies [26,32,33], the hand motor cortex was found to be highly un-isotropic with the lowest rMT found at 45° to the posterior–anterior (P-A) axis and highest rMT for angles perpendicular to the preferred orientation, at 135° and 315° (Figure 4 in [30]). The leg motor cortex is deeper than that of the hand [34,35,36,37]. In order to assess the relative contributions of depth and orientation to the rMT, we now analyzed the ratio of the leg rMT to the hand rMT for eight udTMS orientations of the H7 coil and four rfTMS states covering different 270° portions of the phase space [30]. The ratios were calculated per subject and state. The results are shown in Figure 7.

A repeated measures ANOVA showed a significant condition effect (F (1, 12) = 8.77, *p* = 0.0007). There was a significant difference in rMT ratios among unidirectional orientations (F (1, 8) = 8.71, *p* = 0.0009), with an angle of 45° to the P-A axis yielding the highest rMT ratio, and Tukey post-test revealed a significant difference between the angle of 45° and four of the seven other angles. Angles of 135° and 315° had the lowest ratios, with the leg rMT comparable to the hand rMT (1.10 ± 0.06 and 1.03 ± 0.06, respectively; mean ± SE). Comparison of the results of the four rfTMS polarity states found significant differences (F (1, 4) = 15.37, *p* < 0.0001), with the highest ratio obtained for the state a in Figure 8, where the field vector covers also the angle of 45° to the P-A axis and the two adjacent quarters (see Figure 8a below). Tukey post-test found significant differences between this state and all three other polarity states.

## 5. Discussion

The benefit of rfTMS has been demonstrated in healthy subjects by comparing the physiological effects of rfTMS and conventional unidirectional TMS in the motor cortex. rfTMS induced a significantly lower resting motor threshold (rMT) in both the hand and leg motor cortices compared to unidirectional H7 Coil Deep TMS [29,30]. Moreover, rfTMS led to about 400% higher supra-threshold motor evoked potential (MEP) values in the hand motor cortex and about 700% higher MEP values in the leg motor cortex compared to unidirectional TMS at any orientation [29,30]. The hand motor cortex is known to have high sensitivity to orientation with the preferred orientation for the lowest motor threshold being at 45° to the sagittal plane, perpendicular to the central sulcus [26,32,33]. In contrast, the leg motor cortex is much more isotropic in its organization and has a much lower sensitivity to coil orientation [38,39]. These findings indicate that even in relatively organized networks such as the hand motor cortex, for which the optimal orientation has been experimentally found at 45° to the posterior–anterior axis, rfTMS can recruit neurons in other orientations that induce the desired effect (in this case hand motor activation) at significantly lower induced electric field intensity than that needed for unidirectional TMS at the optimal orientation. Since the hand motor activation with TMS is very sensitive to the accurate coil orientation [26,32,33], the use of rfTMS may also reduce the need for accurate placement and fixation of the coil, and facilitate determination of the motor threshold with procedures such as neuronavigation.

The new analysis presented here on the ratio between leg rMT and hand rMT (following revisiting the data of our recent study [30]) yielded several interesting findings: (1) The highest ratio among udTMS orientations of the H7 coil was found for the angle of 45° to the sagittal plane and was 1.55, similar to the ratio found previously for an H-coil with this orientation [37]. However, much lower ratios were found for other udTMS orientations. In particular, for angles of 135° and 315° to the sagittal plane (which are perpendicular to the preferred orientation of 45°), the rMTs of the much deeper leg TA were comparable to that of the shallower hand APB. This finding emphasizes the importance of orientation and demonstrates that the intensity required to stimulate neurons with unfavorable orientation may be comparable to that required to stimulate much deeper neural structures with less orientational sensitivity. (2) A significantly higher leg/hand rMT ratio was found in the rfTMS state that covers the quarter that includes the most favorable orientation of 45° to the P-A axis, as well as the two adjacent phase space quarters (see Figure S1 in [30] reproduced here as Figure 8). One could expect that the rfTMS state that *does not* cover this favorable quarter would yield a significantly lower ratio compared to the other rfTMS states. However, each of the three rfTMS states that do not cover one of the three quarters covered by the first state yielded ratios with no significant differences between them. This indicates the importance for neural structures with orientations in all these three phase space quarters, for hand motor activation. Although not previously discussed, a similar conclusion may be drawn from the hand rMT results [30], where the three rfTMS states yielded similar rMTs, and the Tukey post-test found significant differences for the first rfTMS state but no significant differences between the three other states.

The delicate motor functions and fine motor skills of the upper limbs include a complex interplay of inhibitory and excitatory inputs [40]. Hence, the combined activation of facilitatory neurons relevant to a certain hand muscle with relevant inhibitory neurons is expected to modulate motor response. Such inhibitory networks may have orientational sensitivity. Therefore, using the rfTMS tool to dissect the contribution of the activation of neurons at the preferred orientation for hand motor activation (e.g., at 45° to the sagittal plane) versus combined activation with various other orientations may reveal such inhibitory networks.

The ability of rfTMS to recruit many neurons in various orientations may enable the achievement of desired physiologic effects at significantly lower intensities compared to conventional unidirectional TMS, thereby potentially reducing undesired side effects due to a reduced field at the scalp and brain surface. On the other hand, the activation of more neurons in various orientations by rfTMS may lead to more side effects and in some cases to the activation of inhibitory networks that may reduce the effect. Future clinical studies will have to address these questions.

All therapeutic clinical trials completed to date with rTMS used commercial unidirectional rTMS devices. The ability of rfTMS to recruit many more neurons in various orientations may enhance the therapeutic effect. As an analogy, a comparison between the effects in MDD of the H1 coil, which stimulates deeper and broader PFC volume, and a figure-8 coil, which stimulates a more focal volume, indicated advantages to the broader stimulation, which recruits more neurons [41]. However, even results where the clinical effect may be reduced with rfTMS would add new information on the mechanisms of action of TMS. Future clinical trials are required to investigate the clinical potential of rfTMS in various brain disorders.

As most clinical treatments with TMS target regions outside the motor cortex, it is important to consider the effects of rfTMS in these locations. MDD is routinely treated with rTMS targeting the dlPFC, which is known to have a very complex morphological organization with high inter-individual variability [27,42]. Many patients with refractory depression do not reach satisfactory improvement with TMS. Computational studies have shown that inter-individual variability in neural orientation optimal for stimulation is larger in the prefrontal cortex (PFC) than in the primary motor cortex [43]. Numerous studies have indicated high variability in the optimal PFC target for the treatment of MDD with TMS [44,45,46]. In addition to the coil location, the orientation of the TMS coil with respect to the underlying cortical anatomy impacts the electric field at the target site [21], and a recent modeling study suggests that orientation accounts for as much variability in the simulated TMS response as does location [47]. Hence, repetitive application of rfTMS, especially with Deep TMS coils, which can induce deeper and broader stimulation [9], will stimulate many more PFC neurons in various orientations, as well as many connections with deeper reward system sites such as the subgenual anterior cingulate cortex (sgACC), ventral tegmental area, and the nucleus accumbens, thus, may give benefit to many MDD patients who do not respond to currently available therapeutic options.

There are numerous potential TMS protocols in the field whose effects are assumed to follow specific dogmas. For example, high-frequency (>5 Hz) TMS is believed to increase neural excitability while low-frequency (1 Hz) is believed to decrease excitability [48]. Intermittent theta burst (iTBS) is expected to increase excitability (facilitation) by inducing potentiation while continuous TBS (cTBS) should decrease excitability (inhibition) by inducing depression [49]. However, there is remarkably high variability in the physiological response, and in many subjects, the modulation of brain tissue excitability is opposite to the expected direction [50,51]. Similarly, there is high variability in the clinical outcomes of each protocol, and surprisingly similar clinical outcomes were found with different stimulation patterns [52] and number of pulses [53,54,55]. Hence, it seems that in many cases inter-individual variability masks differences due to stimulation parameters. Inter-individual differences in morphology may contribute to this variability in TMS effects. rfTMS stimulation of greater numbers of neurons due to reduced sensitivity to orientation may act to reduce this variability and induce more robust therapeutic effects.

Currently, two different protocols are FDA-cleared for the treatment of MDD, high-frequency stimulation, and iTBS [52]. While most of the benefits of rfTMS would be applicable to any protocol, evidence suggests a particular benefit for TBS. It has been suggested [50] that TBS after-effects seem to depend on which neuronal pathways are stimulated. The recruited neural populations can be non-invasively characterized via the onset latencies of motor-evoked potentials (MEPs) elicited by TMS. The types of stimulated neuronal pathways depend to a substantial extent on the TMS orientation. Currents induced at 45° to the anterior–posterior axis (commonly denoted AP) generate indirect waves (I-waves) with longer latency, of neural activity descending the corticospinal tract after a TMS pulse has been applied to the primary motor cortex. In contrast, lateral–medial (LM)-directed currents evoke direct waves (D-waves) featuring the shortest latencies [50,56,57,58]. Hamada et al. [50] found that subjects with a larger difference between the AP and LM latencies (AP-LM) presented the expected after-effects, namely inhibition following cTBS and facilitation following iTBS. In contrast, subjects with shorter AP-LM exhibited the opposite effects. A recent study [59] found that iTBS reduced AP-LM latency difference, and that this reduction significantly correlated with changes in cortical excitability observed following iTBS: subjects with larger reductions in AP-LM latencies had larger increases in cortical excitability following iTBS. This body of literature implies that the recruitment of late I-waves generating pathways plays a vital role in neuroplastic changes induced by TBS, and potentially by rTMS in general. Such pathways exhibit high orientational sensitivity as well as high variability among individuals. Hence, the ability of rfTMS to simultaneously stimulate neurons in various orientations may be a crucial step in enhancing the robustness and magnitude of TMS-induced neuroplastic effects.

TMS is considered to affect the brain to large extent through synaptic plasticity processes of long-term potentiation (LTP), which increases excitability and long-term depression (LTD), which decreases excitability [60]. However, brain network stability requires regulatory mechanisms that oppose these effects and stabilize the system. Metaplasticity refers to changes in the direction or degree of synaptic plasticity based on prior neural activity [60]. In general, homeostatic metaplasticity has a stabilizing effect that maintains the neuronal and network activity within a physiological range [61]. Recent studies indicate that the timing of accelerated TMS sessions may be crucial to the type and amplitude of the neuroplastic after-effects, where with short intervals homeostatic metaplasticity may prevail and eliminate enhancement of the effects of accelerated protocols compared to single session protocols. In contrast, accelerated TMS (e.g., TBS) with longer inter-session intervals (~60 min) led to additive effects and enhanced the neuroplastic changes [62,63]. The rfTMS method opens exciting opportunities to probe the effects of stimulation of various combinations of orientations, as well as all orientations, with different temporal intervals, on the delicate interplay of homeostatic and non-homeostatic metaplasticity mechanisms, and potentially enhance the neuroplastic changes and consequently the clinical impact in various brain disorders.

While applicable to any therapeutic use, rfTMS may be ideally suited for the many brain disorders that are associated with brain networks having diverse orientations. Following stroke, a reorganization of the brain often occurs that may involve reinnervation and axonal sprouting with the formation of new synapses [64]. These processes lead to randomly oriented neural structures in the lesioned brain area. Such structures may be stimulated much more efficiently by rfTMS. Hence, this emerging method has the potential for novel therapeutic options with significantly enhanced efficacy in increasing and facilitating rehabilitation following stroke. In addition, rfTMS potential to generate D-waves may prove beneficial for diagnostic use under anesthesia.

The concept of rfTMS may be extended to use three or more coils and to induce field rotation in three dimensions. For example, an array of two orthogonal coils is placed over the frontal lobe, where one of the coils induces a field along an anterior–posterior axis (*X*-axis), while the second coil induces a field along the lateral–medial axis (*Y*-axis). A third coil can then be positioned over the temporal lobe, inducing an electric field along a superior–inferior axis (*Z*-axis). Importantly, the main field induced by each coil is tangential to the brain surface underneath it, to minimize electrostatic charge accumulation and reduction of the induced field due to a perpendicular field component [65,66,67]. The intensities of each coil are adjusted so that the fields induced by the coils at the target brain region have a similar amplitude. The operation of the three coils may be synchronized. For example, the first coil can be operated with a single biphasic pulse while the second coil can be operated with two consecutive biphasic pulses starting at a time ¼ of a cycle after the first pulse. This leads the field vector to span 270° of the XY plane. The third coil can then be operated with a single biphasic pulse a full cycle after the first coil, corresponding to ¾ cycle after the second coil. Hence, the field vector now spans 360° in the YZ plane. Such a setting will cause the field vector to rotate sequentially in two or more planes, potentially recruiting an even greater number of variably oriented neurons. In another potential implementation, the parameters of rfTMS could be used to deliberately cover less than the full 360° plane in order to stimulate neurons in a specific range of orientations. As another example, two (or more) coils with orthogonal orientations or any other relative angle may be operated with two half-sine pulses by the two coils, consecutively with the onset of the second coil immediately at the end of the first coil with no temporal overlap. This will induce field only in these two orientations. In this way, the differential contribution of neural structures with different orientations can be potentially evaluated with high resolution. This functional information may be combined with anatomical diffusion tensor imaging (DTI) information, resulting in a novel integrated research tool that can increase our understanding of brain circuits, function, architecture, and their inter-relations.

## 6. Conclusions

In the emerging method of rotational field TMS (rfTMS), two orthogonal TMS coils are operated with a 90° phase shift between them and induce a rotating electric field in the brain tissue. This enables the simultaneous activation of many neural structures in various orientations and overcomes the orientation dependence of conventional TMS, which activates only a small fraction of the neurons in a brain target, those aligned in parallel to the induced electric field. rfTMS was shown in healthy subjects to induce significantly stronger physiologic effects in both the hand and leg motor cortices, induce motor activation at significantly lower motor thresholds, and evoke much higher motor-evoked potentials compared to conventional unidirectional TMS [29,30]. Given the activation of a greater number of neurons within a certain brain area, repeated application of rfTMS may induce enhanced neuroplastic effects in neural networks, thus opening novel opportunities for neuroscientific research and for the treatment of neuropsychiatric disorders. rfTMS may be particularly beneficial following stroke, where reorganization of brain tissue with randomly oriented new connections may occur, and be enhanced by effective stimulation. A well-known characteristic of currently available rTMS protocols is significant inter-individual variability, where, in many subjects, effects opposite to those expected are induced. This variability may be at least partially related to orientational sensitivity. Here, rfTMS may have an important role in increasing robustness and enhancing the effectiveness of rTMS protocols.

A fundamental limitation of the technique is that it was tested only in vitro, in vivo, and in healthy subjects. rfTMS is currently being tested in several clinical trials in various brain disorders, including MDD and post-stroke rehabilitation (registered 0034-15 GEH and MOH_2022_02_11_010599 in the Israeli Ministry of Health). Future randomized clinical studies will be necessary to unravel the potential of rfTMS in psychiatric and neurological indications.

## Figures and Tables

**Figure 1 jcm-12-00983-f001:**
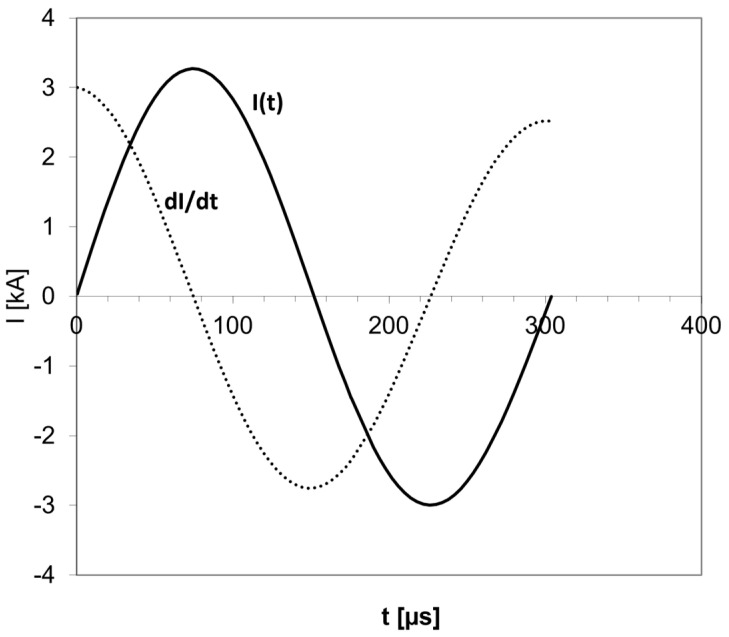
A typical biphasic current pulse in TMS. The current I is in Kiloampere (kA), and the pulse duration is in microseconds. In a biphasic pulse, the current flows in one direction with increasing amplitude, then the amplitude goes down to zero, and the current flows in the other direction. The current I pulse has a sinusoidal shape. The electric field is proportional to the time derivative of the current (dI/dt) and has the shape of a cosine. The brief pulse lasts just a few hundred microseconds.

**Figure 2 jcm-12-00983-f002:**
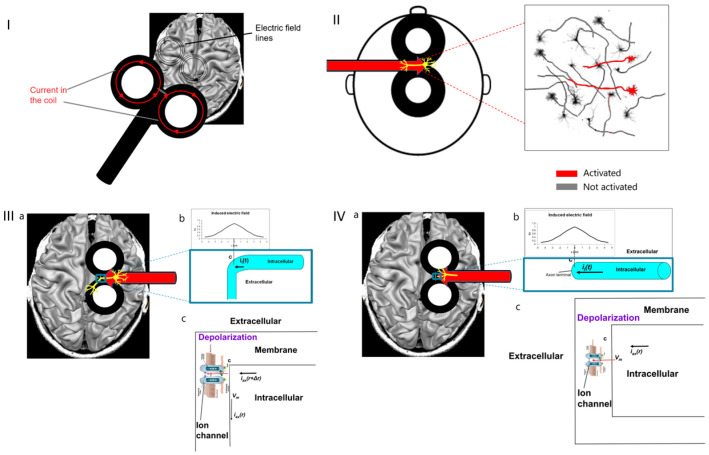
(**I**). Electric field lines induced by a typical figure-8 TMS coil. (**II**). Under the figure-8 central segment, only neurons aligned parallel to the electric field, along the coil axis, are activated (indicated in red). (**III**). Where neurons under the coil center have axon parallel to the induced electric field and bends away from it (**a**), the field is maximal at the bend point (**b**) and leads to transmembrane potential V_m_ across the voltage-gated ion channels (**c**), which are then opened, and action potential is initiated. (**IV**). Where neurons under the coil center have axon parallel to the induced electric field and terminate (**a**), the field is maximal at the axon terminal (**b**) and leads to transmembrane potential V_m_ across the voltage-gated ion channels (**c**), which are then opened, and action potential is initiated.

**Figure 3 jcm-12-00983-f003:**
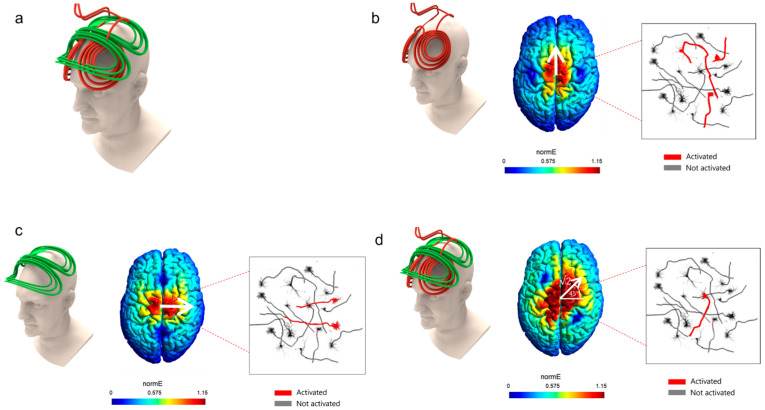
(**a**). Rotational field TMS coils array: two orthogonal coils one on top of the other. (**b**). With only the lower coil operated, electric field is induced along the anterior–posterior axis; thus, only neurons aligned along this axis will be activated (indicated in red). The color code indicates the field intensity (red is highest). (**c**). With only the upper coil operated, an electric field is induced along the lateral–medial axis, thus only neurons aligned along this axis will be activated (indicated in red). (**d**). With both coils operating simultaneously, electric field is induced at 45° to the lateral–medial axis; thus, only neurons aligned along this orientation will be activated (indicated in red).

**Figure 4 jcm-12-00983-f004:**
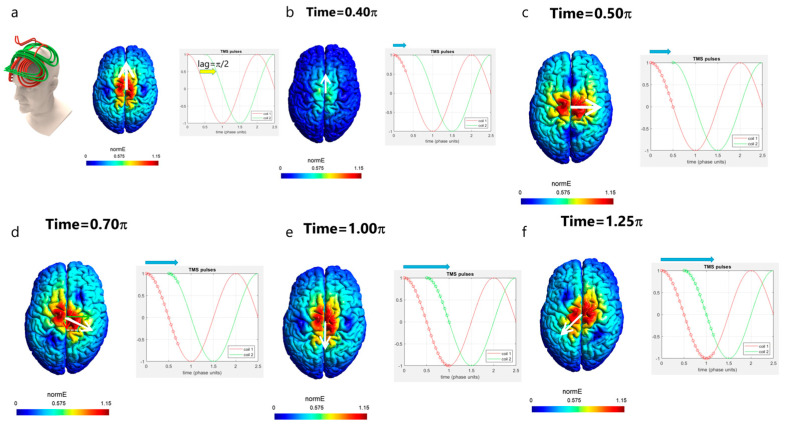
rfTMS scheme: Operation of two orthogonal coils with a lag of ¼ cycle. Shown the evolution of the electric field direction during the pulse in an rfTMS operation. (**a**–**k**) electric field orientation and amplitude at specific timepoints during the pulse, from time = 0 to time = 2.50 π. k. With the rfTMS scheme after completion of the coils’ pulses, neurons in many orientations are activated (indicated in red).

**Figure 5 jcm-12-00983-f005:**
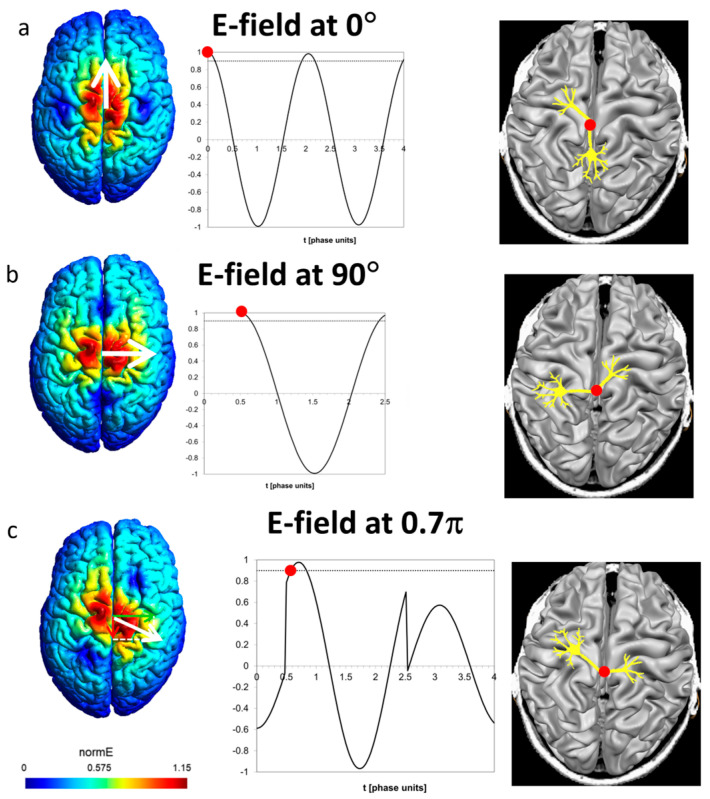
(**a**). Preferred posterior–anterior direction (angle 0°). Only the lower coil contributes to the field along this axis. Stimulation occurs at t = 0. (**b**). Preferred left-right direction (angle 90° = 0.5 π). Only the upper coil contributes to the field along this axis. Stimulation occurs at t = 0.5 π once the lower coil is operated. (**c**). Preferred direction at angle 0.7 π. Both coils contribute to the field along this axis. Stimulation occurs at about t = 0.6 π once the field reaches the threshold (dashed line).

**Figure 6 jcm-12-00983-f006:**
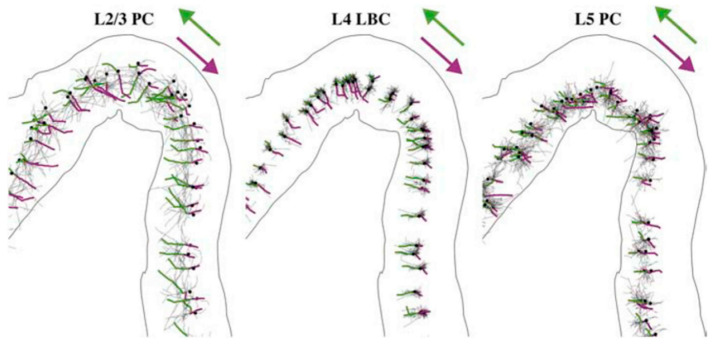
Action potential initiation points in the primary motor cortex layer L2/3 pyramidal cells (L2/3 PC), layer 4 large basket cells (L4 LBC), and layer 5 pyramidal cells (L5 PC) depend on the orientation. Action potentials initiate at terminal points and propagate to proximal branch points. Structures stimulated by monophasic posterior–anterior stimulation are colored in green and by anterior–posterior stimulation by magenta. Somas indicated by black dots. (Reproduced from [22].)

**Figure 7 jcm-12-00983-f007:**
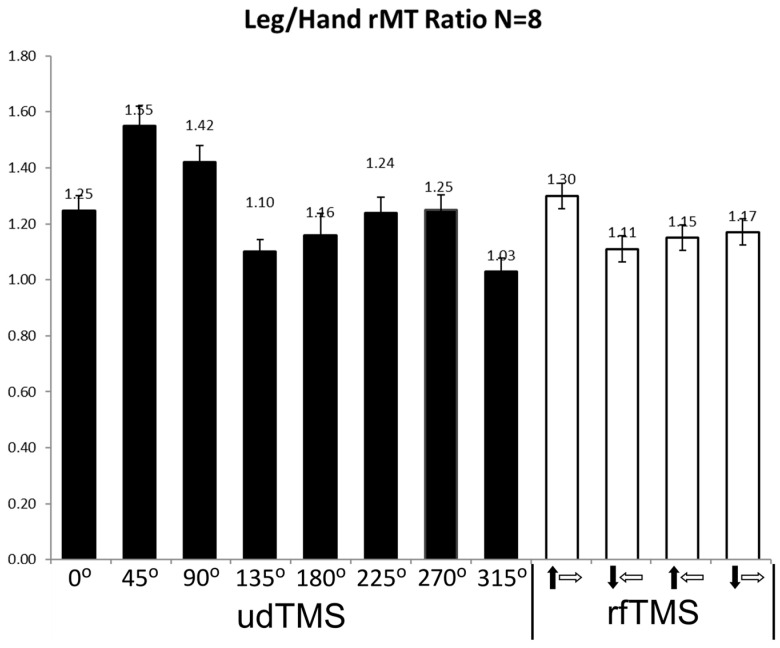
Ratio of leg tibialis anterior (TA) and hand abductor policis brevis (APB) rMTs for various udTMS orientations, and for four rfTMS states. The black and white arrows represent the polarity induced during the second stroke of the biphasic pulse by the lower and upper coils, respectively. Shown are mean ± SE.

**Figure 8 jcm-12-00983-f008:**
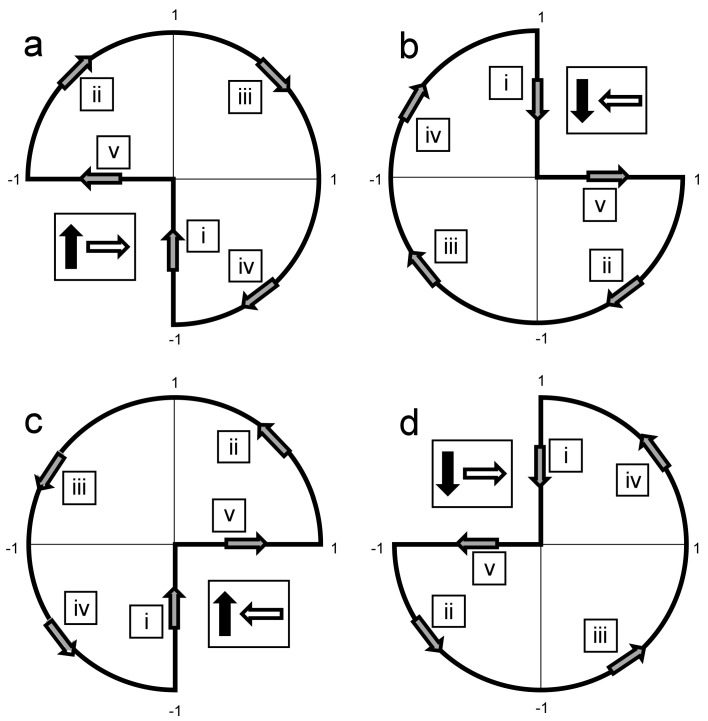
A reconstruction of the effective electric field created from the combined operation of the two perpendicular coils, for the four rfTMS polarity states. The field of coil #1 is directed along the *y*-axis (anterior-posterior) and the field of coil #2 along the *x*-axis (lateral-medial). The effective field completes ¾ of a full cycle during the magnetic pulses, as indicated by the gray arrows. The Latin numbers in squares indicate the order of evolvement of the field vector during the pulses. The polarities were induced such that during the second stroke of the biphasic pulse by the lower and upper coils the induced current in the brain was P-A and L-M (left-right on the left hemisphere) (↑→, panel (**a**)), A-P and M-L (↓←, panel (**b**)), P-A and M-L (↑←, panel (**c**)), or A-P and L-M (↓→, panel (**d**)), respectively. The black and white arrows represent the polarity induced during the second stroke of the biphasic pulse by the lower and upper coils, respectively (Reproduced from [30]).

## Data Availability

Not applicable.

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
