# Peer review of "Revisiting the Rotational Field TMS Method for Neurostimulation"

_jcm, 2023, doi:10.3390/jcm12030983_

Round 1

Reviewer 1 Report

In this article, the authors provide readers with a very clear understanding of rotational field TMS (rfTMS), a variation of traditional TMS that promises to make TMS effects more reliable, especially with regard to treatment effects in clinical populations. Although the content of the article needs to be restructured in many places to follow a red thread, I especially enjoyed reading the section on the outlook within the discussion and am very excited about the ways in which this elegant technique will support neuroscience and patient treatment in the future. The concise video in the supplementary material demonstrating the principles of rfTMS is nothing less than remarkable.

The biggest concern I have is the (in my eyes obvious) mischaracterisation of this article as a "review": only 2 studies are mentioned that fit the narrative (i.e., rfTMS). Also, I wonder why another 2019 study by the authors, "Comparing rotational-field-dTMS to unidirectional-dTMS in healthy volunteers", is not mentioned - I hope I didn't miss it. Given these circumstances and the conciseness of the article, I would consider the "primer" category more appropriate - however, I also cannot attest to any significant contributions to the field over and above the author's 2020 study, "Rotational field TMS: Comparison with conventional TMS based on motor evoked potentials and thresholds in the hand and leg motor cortices", which already contains convincing descriptions of the method as well as empirical data.

Without a doubt, the registered clinical studies indicated in the conclusion will be exciting to follow. Most likely, they will successfully complement and extend the results of the above-mentioned study on the motor cortex and help to revive this almost forgotten TMS approach, which was already conceived in 2014.

Reviewer 2 Report

The authors deal with a relevant and timely topic, i.e., the rotational field transcranial magnetic stimulation TMS (rfTMS) as a novel method for neurostimulation. To this end, they reviewed the literature in order to describe in detail this novel method, its principles, the mode of operation, the effects on the brain, and the potential benefits in increasing the efficacy and decreasing the variability of clinical TMS responses. Overall, the review is nicely conceived and carried out; the studies included seem to be consistent and are adequately discussed. Few comments to the authors, requiring some revision.

Abstract: please provide more details on clinical applications and translational value, as well as on limits.

Introduction: please highlight the added-value of the present review compared to the existing literature.

TMS Principles: please briefly introduced how a standard TMS exam is usually carried out and interpreted.

rfTMS methods: although the narrative design of the review, please include a short “Methods” section including search strategy, databases used, temporal window, inclusion/exclusion criteria, etc.; similarly, a brief “Results” section would be needed, showing the items retrieved, selected, and eventually included. Alternatively, the authors may consider to describe how the relevant studies for this review were chosen.

Discussion: among the neurobiological effects underlying the use of TMS and rfTMS in neurological and psychiatric disorders, please also mention and briefly discuss the possibility to induce and modulate even complex neuroplastic phenomena, such as metaplasticity, as recently reported (e.g., PMID: 34276553).

Conclusions: please summarize them and highlight more the clinical applications and current limits/caveats.

General: although the language is overall acceptable, an editing by a native-English speaker would be useful.

Round 2

Reviewer 1 Report

My main issue unfortunately still holds: I don't see the merit of this work currently, because the main points largely overlap with the author's 2020 paper, "Rotational field TMS: Comparison with conventional TMS based on motor evoked potentials and thresholds in the hand and leg motor cortices". Also, the title of the manuscript would need adjustment given that the method is not new in a chronological sense: the basic principle was established in the mentioned 2014 study already.

I would be happy to reconsider this point when more empirical work can be integrated in a review, especially the hinted-at clinical trials.

Author Response

My main issue unfortunately still holds: I don't see the merit of this work currently, because the main points largely overlap with the author's 2020 paper, "Rotational field TMS: Comparison with conventional TMS based on motor evoked potentials and thresholds in the hand and leg motor cortices".

Response

We thank the reviewer for the comment and have now clarified that this is not a classical review paper but mainly a perspective of an emerging technology which includes revisiting previously published data.  In addition to providing a detailed description of this emerging technology based on recent literature, we have now revisited data from the 2020 paper and performed additional analyses shedding interesting light on the relative importance of orientational sensitivity and depth in TMS.  This new analysis emphasizes the potential importance of rfTMS ability to stimulate neurons in all as well as in selected orientations as demonstrated by calculating for each subject the ratio between intensities required to stimulate the hand and leg (deeper) motor areas, using the various orientations or the rfTMS approach. We believe that providing a detailed perspective as well as new visualizations, including an animation movie demonstrating the technique and expanded discussion of an emerging technology merits publication and will benefit the readers.

Also, the title of the manuscript would need adjustment given that the method is not new in a chronological sense: the basic principle was established in the mentioned 2014 study already.

I would be happy to reconsider this point when more empirical work can be integrated in a review, especially the hinted-at clinical trials.

Response

We absolutely agree with the reviewer and have now revised the title accordingly.  In addition, the credit to the original work published in 2014 is clearly provided in the current paper. The corrected title is: “Revisiting the Rotational field TMS method for neurostimulation”
